# Nitrogenous Bases in Relation to the Colloidal Silver Phase: Adsorption Kinetic, and Morphology Investigation

**Malgorzata Zienkiewicz-Strzalka and Magdalena Blachnio \***

Faculty of Chemistry, Maria Curie-Sklodowska University, M. Curie-Sklodowska Sq. 3, 20-031 Lublin, Poland; malgorzata.zienkiewicz-strzalka@mail.umcs.pl
\* Correspondence: magdalena.blachnio@mail.umcs.pl; Tel.: +48-0815375637

**Abstract:** The interaction between inorganic nanoparticles and biological molecules is of great importance in the field of biosystems and nanomaterials. Here, we report the adsorption process of a heterocyclic organic compound (nitrogenous base) on a microporous carbon (C) in the presence of a colloidal silver solution (AgNP solution) as an accompanying substance. Analysis of the potential colloid–biomolecule interaction as well as the subsequent phenomenon of changes in the morphology of the colloidal system in the presence of selected nucleotides was investigated. Adenosine nitrogenous base (Anb) was selected as a model molecule of the building block of DNA and RNA. The adsorption process of nucleotides from one- and two-component systems was monitored by cyclic UV-VIS measurements for obtaining time-dependent profiles and estimating the kinetic characteristics of uptake. We demonstrate the temperature-dependent course of the adsorption process with visible nucleotide-AgNP morphology determinants. The experimental adsorption kinetics were analyzed using selected theoretical models (intraparticle diffusion model, multiexponential equation, and many others). On the other hand, obtained Anb/C and Anb/AgNP/C composites were characterized by various techniques suitable for material surface and morphology characterization: high-resolution transmission electron microscopy (HR-TEM and TEM/EDX), $N_2$ physisorption measurements, and thermal analysis (thermogravimetric analysis (TGA)/differential scanning calorimetry (DSC) experiments).

**Keywords:** nitrogenous bases; silver nanoparticles; adsorption; kinetics; multicomponent system

## 1. Introduction

The importance of nanosized metal clusters is well known and appreciated in various sectors of science and modern technology. Due to the rapid development of nanotechnology, their products are becoming increasingly common in almost all areas of our life. Noble metal nanoparticles are meaningful components in a wide area of applications, technological solutions, and future developments due to their physical and chemical properties correlated with size and morphology [1–3]. Based on nanoscale elements, the nanocomposite materials, as well as other hybrid structures, are designed and produced on a fairly large scale. The idea of immobilization of the nanoparticles on solid surfaces may lead to their stabilization and avoid uncontrolled release to nature due to the increasing problem of their safety on the health of living organisms and the natural environment [4,5]. These concerns are associated with the potential impact of nanosized particles on living organisms and cells, and thereby on the building blocks of genetic information, their replication, and structural construction [6–8]. Interactions between DNA, proteins, and many other types of bioactive molecules have been an important area of research due to the possible interaction of silver nanoparticles with different binding sites in the DNA chain, including nucleobases, sugars, and phosphates [9]. Based on this, heterocyclic molecules containing phosphorus and nitrogen elements are important due to their applicability in the regulation of enzymatic activity and interaction with biological systems [10]. A comprehensive approach to the issue of

phosphorus nitrogen heterocycles and their importance is presented in the literature [11]. It is worth paying attention to the modification of the surface of solid carriers in the form of metal oxides with organic phosphonic acid derivatives, which are an excellent example of the construction of organic–inorganic hybrid materials [12]. Among them, heterocycle systems (with phosphorus and nitrogen elements) are valuable for their importance in biological systems [13,14]. For example, an adenosine molecule can be presented, whose derivatives, i.e., (i) deoxyadenosine is a building block of DNA, (ii) adenosine mono-, di-, or triphosphate (AMP/ADP/ATP) act as energy carriers [15], and (iii) cyclic adenosine monophosphate (cAMP) is pervasive in signal transmission [16]. The role of phosphorus in the living cell comes with a few features. This element is part of cellular membranes as building phospholipids, it is a major element of DNA and RNA nucleic acids [17], and it is a key nutrient involved in the transfer of energy and the synthesis of several cellular components (ATP and NADPH), involved also in the photosynthesis process. The adsorption of heterocyclic compounds seems to be very important in the context of the functioning of biological systems and chemical processes. In the first case, the adsorption of heterocyclic compounds in the biological system can be represented as a surfactant–polyelectrolyte assembly. Interactions between polyelectrolytes and surfactants can be studied in the bulk phase and on the solid surface. The method used to determine the interaction is the measurement of the surface tension curve of the surfactant in the presence and absence of the polyelectrolyte [18–20]. On the other hand, possible mechanisms of interaction may include noncovalent interaction between the negative charge of biomolecules and metal ions. One of the examples is the interaction between the negative charge of the phosphate backbone. Silver nanoparticles exhibit a high affinity to sulfur and nitrogen atoms and their derivatives [21]. The affinity phenomena of silver nanoparticles towards sulfur and nitrogen atoms was investigated in detail in the literature [22]. The aggregation effect of the particles, visible change of colors, and change in the surface plasmon resonance characteristics [23] indicate the possible interaction between metal clusters and biomolecules. The optical properties of silver nanoparticles are related to the localized plasmon resonance which, as an unusual phenomenon, is reserved only for a small group of nanostructures. Due to it, control of their properties that facilitates research is possible [3,24–27].

On the other hand, the specific interaction between nanoparticles and biomolecules is willingly used in practice. Modern systems and materials containing metallic nanoparticles and biomolecules such as components of genetic information in the form of a monolayer or thin films can be used as accurate diagnostic sensors. The phenomenon of interaction between biomolecules and metal nanoparticles seems to be an interesting problem and covers at least several scientific disciplines such as the chemical and biological sciences. The adsorption process of organic molecules on surfaces of nanostructured solids and constructions with metal nanostructures is significant when systems have to be applied as biosensors [28–30], bionanotechnology systems, and biocatalysts. These applications are, many times, based on the phenomenon of surface-enhanced Raman spectroscopy (SERS), surface plasmon resonance spectroscopy (SPRS), and electrochemical and calorimetric detection methods [31–33]. On the contrary, specified biosensors may contain accurate molecules, such as proteins, genes, and their derivatives such as small segments of deoxyribonucleic or ribonucleic acid. Whereas oligonucleotides attach to their target, they generate a signal which is possible to measure. As follows, this type of biosensor can be used to detect specified genes and building blocks of genes related to viral infections, cancer, and other diseases. However, such advanced solutions require elementary studies on the synthesis and characterization of biomolecule–nanoparticle composites. The adsorption of organic molecules with biological relevance is still a great challenge due to their large sizes and complex structures.

The goal of this work was to adsorb adenosine nitrogenous on a surface of porous activated carbon in the presence of silver nanoparticles as an accompanying substance. The adsorption process of a heterocyclic organic compound (nitrogenous base) was considered in terms of kinetic description. The interactions of the components with solid substrates, as

well as the influence of the presence of a nitrogenous base on the morphology and size of silver nanoparticles, were considered and evaluated.

## 2. Materials and Methods

### 2.1. Materials

Nitrogenous base (adenosine, Anb) (Figure 1), silver nitrate (>99%), sodium citrate (citric acid trisodium salt dehydrate), polyvinylpyrrolidone (PVP), and sodium borohydride (NaBH$_4$) were supplied by Sigma-Aldrich. Two last reagents were used as a polymeric stabilizer of silver nanoparticles and a reductant of silver ions, respectively. All solutions were prepared with deionized water. Reagents were used without further purification. The highly porous activated carbon RIAA was purchased from Norit n.v., Amersfoort, Netherlands, and applied as a carrier material for adsorption experiments.

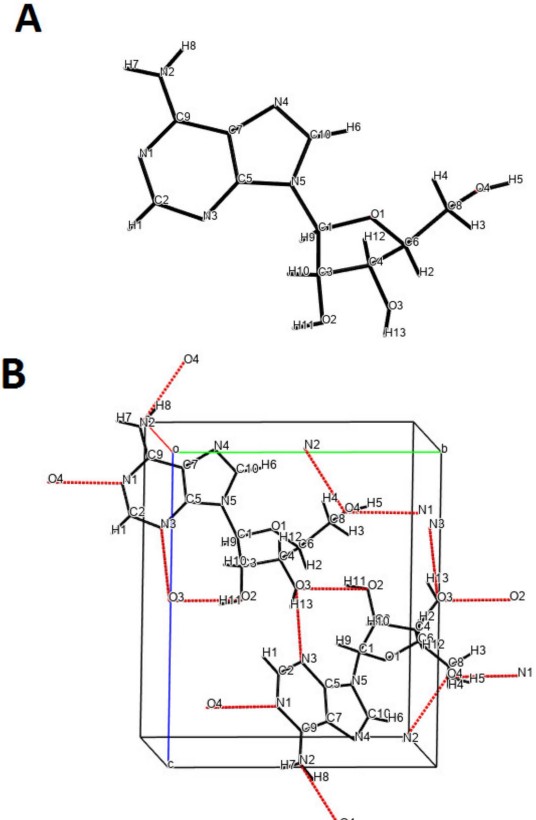

**Figure 1.** (**A**) The structure of the adenosine nitrogenous base (in crystal phase) and (**B**) the unit cell packing with potential H-bond contacts (red lines).

### 2.2. Preparation of Colloidal Silver Solution

Silver nanoparticles solution was obtained by the reduction of silver nitrate by NaBH$_4$ in the presence of poly(vinyl pyrrolidone) (PVP) and sodium citrate as a stabilizer. In this step, 50 mL of distilled water and 0.1 mL of 1 M aqueous sodium citrate solution were mixed followed by the addition of 0.1 mL of 5% aqueous AgNO$_3$ solution. After 10 min stirring, 10 mg of NaBH$_4$ was added to the solution to reduce the silver ions. As a result, the transparent yellow-orange solution of colloidal silver was obtained. The color of the solution indicates the presence of small silver nanoparticles. A performed synthesis procedure allowed to get a stable system without visible effects of nanoparticles agglomeration.

### 2.3. Preparation of Anb/AgNP/C Composite

Nitrogenous base/carbon and nitrogenous base/colloidal silver/carbon composites were prepared by deposition of a specified amount of Anb and AgNP onto the activated

carbon by 24-h incubation at 25 °C. Typically, 0.5 g of AC was suspended in 20 mL of a water solution containing 100 μL of Anb for nitrogenous base/carbon composite. Similarly, 0.5 g of AC was suspended in 20 mL of a water solution containing 100 μL of Anb and 100 μL of AgNP for nitrogenous base/colloidal silver/carbon composites. The mixtures were placed for 24 h in the shaker in isothermal conditions. The resultant samples were rinsed with distilled water and gently dried in an air atmosphere at room temperature. After these steps, the Anb/C and Anb/AgNP/C composites were obtained.

### 2.4. Methods and Calculations

#### 2.4.1. Porosity Evaluation by Low-Temperature Nitrogen Sorption

Structural characterization and evaluation of the porosity of the samples before and after the introduction of nitrogenous bases as well as silver nanoparticles were performed using a low-temperature isothermal nitrogen adsorption–desorption using an automatic Micromeritics device (ASAP2020). Specific surface areas were computed from experimental isotherms by applying the BET theory and linear range of the BET plot [34]. Pore size distribution functions (PSD) of samples were calculated using the Horvath-Kawazoe (HK) method (slit geometry) [35,36] and the nonlocal density functional theory (NLDFT) method [37] using non-negative regularization of 0.001. In this case, the slit geometry of pores was chosen as a model of carbon porosity. For comparison, calculations of pore size distributions following the Barrett, Joyner, and Halenda (BJH) procedure [38] were applied. All samples were outgassed before analysis at 100 °C for 24 h in degas port of the analyzer. The dead space volume was measured for calibration before experimental measurements of the samples using helium as an adsorbate.

#### 2.4.2. Adsorption Studies by Kinetic Approach

Adsorption kinetic measurements were carried out based on complete spectra in the UV-Vis range using a UV-Vis spectrophotometer (Cary100, Varian, Melbourne, Victoria, Australia) equipped with a flow cell allowing the cyclic solution measurements in a closed system. After every step of measurement, the solutions were automatically returned to the reaction vessel. The ~0.1 g of the carbon sample was loaded into a quartz temperature-controlled vessel connected with a stirrer (EUROSTAR 20, IKA, Poznan, Poland) (120 rpm) and thermostat (Ecoline RE207, Lauda, Germany). At definite time intervals, the solutions were gathered in a flow cell and the absorption spectra were measured in the wavelength range of 200–700 nm. Kinetic measurements were carried out at three temperatures: 10 °C, 25 °C, and 40 °C. Before the kinetic experiment, the blank, i.e., the zero-time solution, was prepared without carbon adsorbent, and the full UV–vis spectrum was registered. The carbon sample was degassed under a vacuum (100 °C for 24 h) to dry and get rid of unnecessary volatile impurities in the sample. The experimental kinetic data were optimized using selected adsorption kinetic models and theories presented in Table 1.

**Table 1.** Kinetic equations and models of adsorption data.

| Kinetic Equation (Abbreviation) | Short Description | Kinetic Equations |
|---|---|---|
| First-order reaction (FOE) | Adsorption kinetics in the systems with a standard concentration gradient and processes based on diffusion, (without intraparticle diffusion) [39]. | General form : $\frac{dc}{dt} = -k_1(c - c_{eq})$; (1) <br> Linear form : $ln(c_{eq} - c) = ln(c_{eq} - c_o) - k_1t$ <br> where $c$ is the temporary concentration, the $o$ and $eq$ are initial and equilibrium states respectively, and $k_1$ is the adsorption rate coefficient [40,41]. |
| Pseudo-first order (PFOE) | Adsorption kinetics is dependent on adsorption value instead concentration [42]. | General form : $\frac{da}{dt} = k_1(a - a_{eq})$ (2) <br> Linear form: $ln(a_{eq} - a) = lna_{eq} - k_1t$ <br> where $a$ is the actual adsorbed amount. |

**Table 1.** *Cont.*

| Kinetic Equation (Abbreviation) | Short Description | Kinetic Equations |
|---|---|---|
| Second-order equation and pseudo-second-order equation (SOE and PSOE) | Suitable when concentration changes proceed rapidly and the rate is proportional to $(a_{eq} - a)^2$ [43,44]. | $a = a_{eq}[k_2 t / (1 + k_2 t)]$ (3) or $t/a = (1/a_{eq})(1/k_2 + t)$ and $a = a_{eq}[k_2 t / (1 + k_2 t)]$ where $k_2 = k_{2a} \cdot a_{eq}$ and $k_{2a}$ are the rate coefficients for pseudo-second-order kinetics [43]. Pseudo-second-order equations may be used for energetically heterogeneous surfaces. |
| 1,2-mixed-order equation (MOE) | Generalization of the first- and second-order kinetic equations. Relevant for the systems which behave in the middle of these two [45]. | $F = a/a_{eq} = \frac{1 - exp(-k_1 t)}{1 - f_2 exp(-k_1 t)}$ or $ln\left(\frac{1-F}{1-f_2 F}\right) = -k_1 t$ (4) where $F$ is the relative adsorption progress in time, $f_2 < 1$ is the normalized share of the second-order process in the kinetics. In special cases, the MOE equation may be degraded to the simple kinetic equations of the first ($f_2 = 0$) and the second order ($f_2 = 1$) type [46,47]. |
| Fractal-like MOE equation (f-MOE) | Includes probably distribution of rate coefficients. Adequate description of the non-ideality effects [48]. | $F = \frac{1 - exp(-k_1 t)^p}{1 - f_2 exp(-k_1 t)^p}$ (5) where $p$ is the fractal coefficient. |
| Multiexponential equation (m-exp) | Generalization of Lagergren equation to a series of parallel first-order processes [49,50]. Right for a description of the adsorption kinetics on energetically heterogeneous solids that could not be expressed by the FOE/SOE/MOE equations [51]. | $c = (c_o - c_{eq}) \sum_{i=1}^{n} f_i exp(-k_i t) + c_{eq}$ (6) or $c = c_0 - c_o u_{eq} \sum_{i=1}^{n} f_i [1 - exp(-k_i t)]$ where "$i$" is the term of m-exp equation, $k_i$ is the rate coefficient and $u_{eq} = 1 - c_{eq}/c_o$ is the relative loss of adsorbate from the solution. |
| Intraparticle Diffusion Model (IDM, Crank) | Adsorption on the spherical adsorbent grains [52]. | $F = 1 - \frac{6}{\pi^2} \sum_{n=1}^{\infty} \frac{1}{n^2} exp\left(\frac{-\pi^2 \cdot n^2 \cdot D_a \cdot t}{r^2}\right)$ (7) where $r$ is the radius of the adsorbent particle, $D_a$ is the effective diffusion coefficient: $D_a = \frac{D}{\tau_p \cdot (1 + \rho \cdot K_H \cdot \varepsilon_p)}$ (8) where $D$ is the molecular diffusion coefficient, $\tau_p$–the dimensionless pore tortuosity factor, $\rho$ is the particle density, $\varepsilon_p$ is the particle porosity, $K_H$–the Henry adsorption constant. |
| Pore Diffusion Model (PDM, McKay) | Also called as shrinking core approach. Adsorption on porous solids (additional resistance connected to the transition through the solution/particle interfacial layer) [53]. | $\frac{dF}{d\tau_s} = \frac{3(1 - u_{eq} \cdot F) \cdot (1 - F)^{\frac{1}{3}}}{1 - B \cdot (1 - F)^{\frac{1}{3}}}$ (9) where $u_{eq}$ is the relative adsorbate loss, the parameter $B = 1 - 1/B_i$, where $B_i = K_f/D_p$ is the Biot number, $D_p$ is the pore diffusion coefficient, $K_f$ is the external mass transfer coefficient, $\tau_s$ is the undersized model time: $\tau_s = \frac{1}{6 \cdot u_{eq}} \left\{ \left(2B - \frac{1}{b}\right) \cdot ln\left[\left\vert\frac{x^3 + X^3}{1 + X^3}\right\vert\right] + \frac{3}{a} ln\left[\vert\frac{x + X}{1 + X}\vert\right] \right\} + \left\{ arc \tan\left(\frac{2 - X}{X \cdot \sqrt{3}}\right) \right\} - arc \tan\left(\frac{2 \cdot x - X}{X \cdot \sqrt{3}}\right)$ (10) where $x = (1 - F)^{\frac{1}{3}}$, $b = \left(\frac{1}{1 - u_{eq}}\right)^{\frac{1}{3}}$ (11) |

### 2.4.3. Other Techniques and Calculations

Transmission electron microscopy with energy-dispersive X-ray analysis (TEM/EDX) was carried out on a high-resolution scanning transmission electron microscope Titan G2 60–300 (FEI, Hillsboro, OR, USA). A thermal characteristic was carried out on a simultaneous thermal analyzer NETZSCH STA 449 F3 Jupiter (Netzsch, Germany, 2013) with a steel furnace in the temperature range of 30–950 °C. A thermal analysis of each sample (mass ~20 mg) was performed under a synthetic air atmosphere (25 mL·min$^{-1}$) with a heating rate of 20 °C·min$^{-1}$. A TG/DSC analysis was carried out in an atmosphere of synthetic air (80:20 of $N_2$:$O_2$) as ultrahigh purity gases from Zero Air Plus (6.0) (Air Products and Chemicals, Inc., Allentown, PE, USA). The microelectrophoretic mobility of the silver colloid solution, as well as particle size measurements, were performed by the Zeta Pals Bi–Mas device (Brookhaven Instruments, Holtsville, NY, USA). AAS-3 (Carl Zeiss, Jena,

Germany) atomic absorption spectrometer was applied for silver content determination in the initial colloidal silver nanoparticles solution.

## 3. Results and Discussion

### 3.1. Characteristics of Silver Colloidal Solution

The electrophoretic mobility (zeta potential) of an AgNP solution allows for a description of the electrical interaction forces between the dispersed particles in colloid solutions and the characterization of the silver nanoparticle system in regard to all compounds in the solution. The stabilization effect (resultant from electrostatic and steric stabilization) was estimated by measuring their electrophoretic mobility values for a fresh-prepared solution, as well as for the same solution after one month. The zeta potential values were found to be quite low, in the range of $-3 \times 10^{-1}$ mV to $-7 \times 10^{-1}$ mV without significant changes over a month. The silver concentration was determined as 43.5 mg/L by atomic absorption spectrometry. The spectroscopic characterization of the AgNP solution is shown in Figure 2. UV-visible spectra indicate the presence of AgNP in the solution phase with $\lambda_{max}$= 400 nm (changes in the solution color from colorless (typical for the solution of silver precursor) to yellow-brown (typical for the silver nanophase)).The low half-width of the plasmonic peak and the lack of significant differences in its width for one month from the moment of synthesis suggest a stable system with low polydispersity.

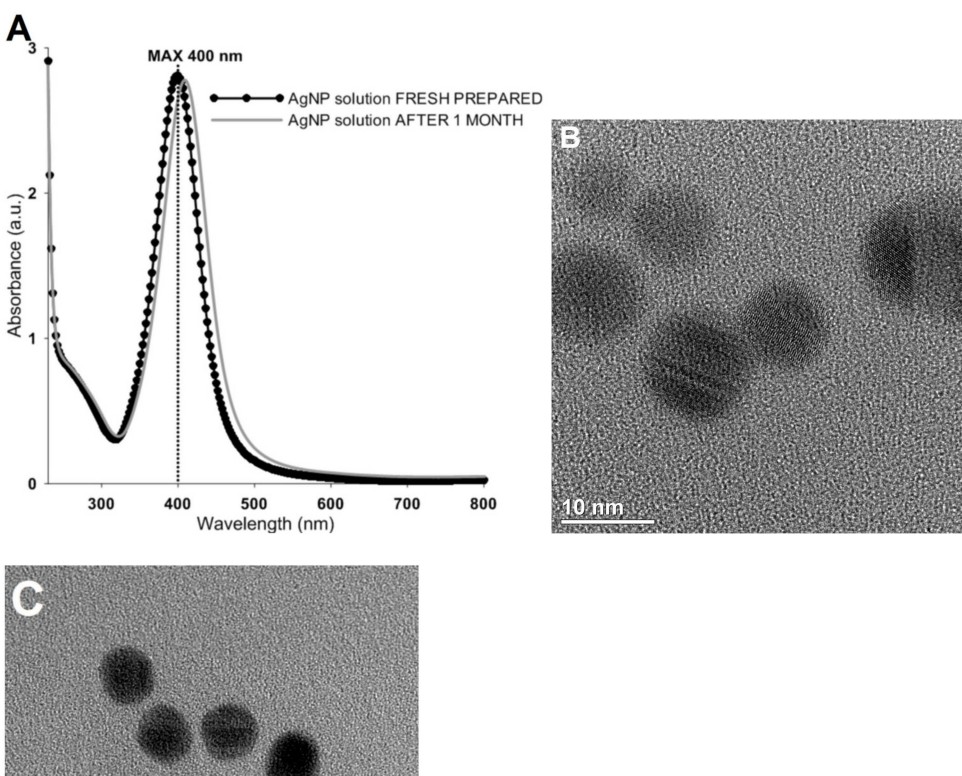

**Figure 2.** (**A**) UV-Visible Spectra of as-synthesized silver nanoparticles and the same solution after one month and (**B**,**C**) the corresponding TEM images of AgNP in AgNP solution.

### 3.2. Characteristics of Anb/C and Anb/AgNP/C Composites

$N_2$ adsorption–desorption analysis at 77 K was performed for activated carbon samples before and after the incorporation of nitrogenous bases and silver nanoparticles. The textural properties determined from the experimental isotherms are summarized in Table 2.

**Table 2.** The values of textural parameters of the porous structure of AC and composite samples.

| Sample | Surface Area ($S_{BET}$) (m²/g) | | Pore Volume (cm³/g) | | Pore Size Distribution (nm) | | |
|---|---|---|---|---|---|---|---|
| | $S_{Total}$ [1] | $S_{mic}$ [2] | $V_{Total}$ [3] | $V_{mic}$ [4] | $D_h$ [5] | $D_{mo(NLDFT)}$ [6] | $D_{mo(HK)}$ [7] |
| AC | 1441 | 1071 | 0.80 | 0.49 | 2.22 | 0.54 | 0.52 |
| Anb/C | 1234 | 666 | 0.72 | 0.35 | 2.33 | 0.61 | 0.55 |
| Anb/AgNP/C | 1207 | 535 | 0.69 | 0.29 | 2.29 | 0.71 | 0.56 |

[1] BET surface area calculated using experimental points at a relative pressure of ($P/P_0$) 0.035–0.31 where p and $p_0$ are denoted as the equilibrium and saturation pressure of nitrogen; [2] Surface area, [3] Total pore volume calculated by 0.0015468 amount of nitrogen adsorbed at $p/p_0 = 0.99$, [4] Pore volume of micropores calculated by t-plot method with fitted statistical thickness in the range of 3.56 to 4.86 Å, [5] Hydraulic pore diameters calculated from the BET surface areas and pore volumes according to the equation: $D_h = 4\,V/S$; [6] The pore diameter estimated from PSD maximum of nonlocal density functional theory; [7] The pore diameter estimated from PSD maximum of Horvath-Kawazoe theory.

The $N_2$ adsorption–desorption isotherms for investigated systems, as well as pore size distributions (PSD) from desorption data, are presented in Figure 3. Overall, the obtained isotherms are similar in shape for all investigated samples. A strong increase in the adsorbed gas quantity in the initial range of isotherms (relative pressure below 0.03) indicates the existence of a significant amount of micropores. In this instance, the quantities of adsorbed nitrogen at 0.03 $p/p_0$ are about 260 cm³/g (STP). The significant amount of micropores even for a sample containing both Anb molecules and AgNP (Anb/AgNP/C) may suggest that micropores are not blocked by adsorbates and allow to maintain good textural properties of the carbon support.

Adsorption–desorption isotherms show typical H4 hysteresis loops according to slit-shaped pores. The course of adsorption isotherms suggests a combination of two types of isotherms (I and II type) rather than a well-defined single type. The shape and existence of the hysteresis loop indicate two distinct types of porosity (micro and mesopores). Figure 3B–D shows micropore-size distributions of initial and modified carbon samples. The micropore size distribution curves suggest that small micropores (below 1 nm) cover a substantial contribution in the micropores range. In all cases, the PSD functions start at very similar pores, and the first maximum according to the HK method appears at 0.5 nm. Here, similar sequences of points on the PSD curve connected with HK and NLDFT theories are observed.

When BJH theory is applied for pores distribution characterization, the PSD of mesopores can be generated. Obtained PSD curves suggest that all investigated samples contained mesopores with sizes of 4 nm. In greater detail, the specific surface area of the unmodified carbon adsorbent was determined as $S_{total}$=1441 m²/g with total pore volume $V_{total}$=0.80 cm³/g according to the nitrogen amount adsorbed at a relative pressure $p/p_0$ of 0.99. Activated carbon RIAA was found to be highly microporous, with more than half of the pores falling on micropores. The specific surface area of micropores was impressive and equals $S_{mic}$=1071 m²/g with a micropore volume of about $V_{mic}$=0.49 cm³/g. After the deposition of the silver nanoparticles and further adsorption of the nitrogenous base, significant changes in the porous structure of activated carbon were observed. It should be emphasized the significant reduction of specific surface area and availability of micropores due to their blocking by larger particles of silver nanoparticles and adsorbate molecules. After the deposition of Anb and AgNP, the specific BET surface area decreased by 207 m²/g and by 234 m²/g for Anb/C and Anb/AgNP/C samples respectively. The surface areas of

micropores and pore volume were also significantly lower (by 405 $m^2$/g and pore volume lower by 0.14 $cm^3$/g and 536 $m^2$/g and pore volume lower by 0.20 $cm^3$/g for both samples).

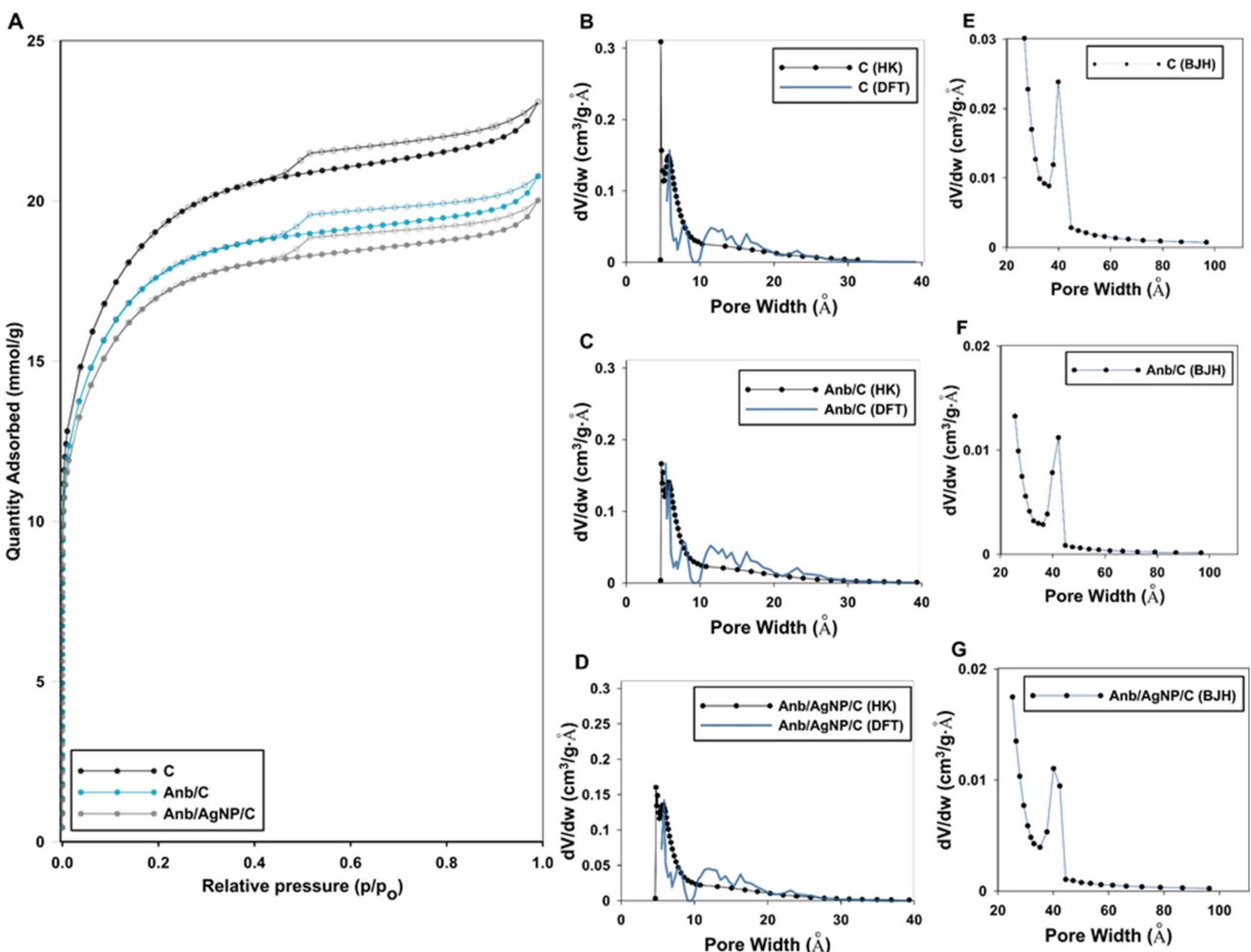

**Figure 3.** (**A**) Nitrogen adsorption–desorption isotherms plots at 77 K of investigated samples; (**B**–**D**) Porosity distributions by Horvath-Kawazoe (differential pore volume plots) and Nonlocal Density Functional Theory (NLDFT); (**E**–**G**) Porosity distributions by BJH theory.

Thermal analysis of final products was performed to assess their thermal stability before and after the adsorption of nitrogenous bases. Figure 4 shows the results of thermal analysis represented by the first derivative of the thermogravimetric curves (DTG in Figure 4A), mass loss versus temperature (TG curves in Figure 4B), and differential scanning calorimetry (DSC curves in Figure 4C) for the single nitrogenous base, nitrogenous base/carbon material, and AgNP modified nitrogenous bases/carbon material. The initial carbon carrier remains stable up to approximately 450 °C, as evidenced by the lack of significant changes in the temperature range below this point. In the range of 400–950 °C, the exothermic surface oxidation and pyrolysis process were observed. The total mass loss up to 800 °C for the initial carbon sample was specified as 86%. On this basis, the carbon samples loaded with Anb and AgNP were also analyzed. Higher mass losses were observed for all modified samples as well as for individual Anb.

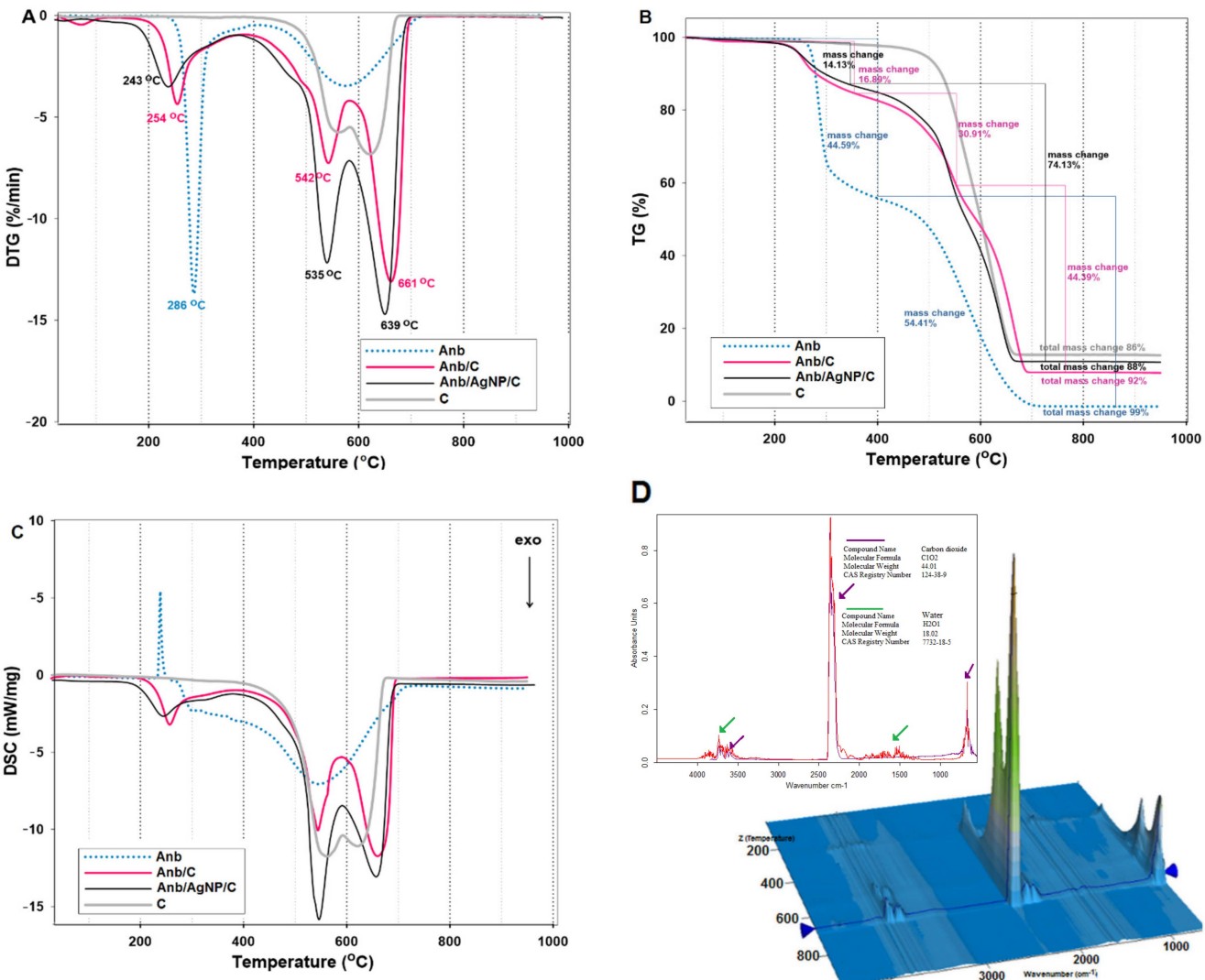

**Figure 4.** Comparison of DTG (**A**), TG (**B**), and DSC (**C**) curves measured for C, Anb/C, Anb/AgNP/C, and individual nitrogenous base (Anb) samples. (**D**) The 3D TG-IR spectra of the gas phase of decomposition products during thermal treatment at 600 °C for Anb/AgNP/C.

For the individual nitrogenous base, the first step includes the temperature range from 200 °C to 300 °C (endothermic peak at 286 °C) and the additional DTG exopeak observed at 550 °C. After the adsorption of nitrogenous bases and a further AgNP, the shift of the initial signal to the lower temperature was observed. The ΔT was described as 32 °C and the next 21 °C for Anb/C and Anb/AgNP/C samples, respectively, and indicates a decrease in the thermal stability of the nitrogenous base after binding it in the form of an adsorbed system. The occurrence of mass loss due to carbon dioxide and water release (IR spectra, Figure 4D) was confirmed during an evolved gas analysis of Anb/AgNP/C (by FTIR).

The contact of Anb with AgNP causes the color change of the colloidal silver solution from orange-yellow to a red color almost immediately. The TEM analysis of initial (AgNP) and mixed (AgNP+Anb) solutions confirmed that this phenomenon is associated with a change in the size and morphology of silver nanoparticles from the most spherical (Figure 5A–C) to more complex structures with triangular and hexagonal morphology. The solution of Anb/AgNP contains silver in the form of hexagonal morphology with a size range of 10 to 20 nm (Figure 5D–F).

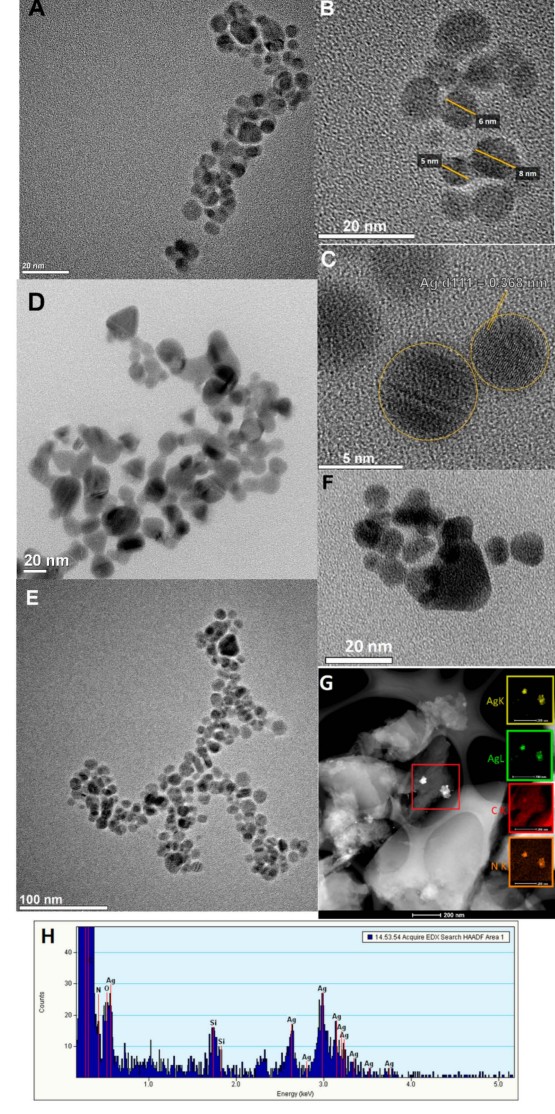

**Figure 5.** TEM images of the initial solution of silver nanoparticles (**A–C**); silver nanoparticles after contact with adenosine (**D–F**), (**G**) STEM as dark field image of the Anb/AgNP/C sample separated from the solution after adsorption with EDX mapping images for Ag, C, and N elements and (**H**) EDS spectrum of the investigated sample from the area marked as a red square.

TEM measurements have been supplemented with EDX measurements as a tool for measuring the energy and the number of characteristic X-rays generated during electron beam irradiation of the selected area of the samples. In this case, such an analysis was performed on a carbon sample separated from the Anb/AgNP/C adsorption system. Here, elemental mapping obtained by EDX using the STEM mode was applied for illustrating the distribution of Ag (K), Ag (L), C (K), and N (K) elements. It was found that great elements localized on the support surface constitute the silver nanoparticles in the agglomerated form. Moreover, N (K) distribution was fairly homogeneous distributions both in the area of the carbon carrier and within the visible silver phase. This allows us to assume that adenosine locates equally readily on the carbon phase and maintains a good affinity to the silver phase. The presence of the components was also significantly confirmed by the EDX spectrum in the range of energy from 0 to 5 keV (Figure 5H).

In this section, the process of adsorption of the nitrogen base adenosine on microporous carbon in the presence of a colloidal silver solution as an accompanying substance was investigated according to the kinetic approach. However, it should be noted that the liquid–solid sorption process consists of several steps: adsorbate mass transfer by diffusion from

the bulk solution to the outer surface of the solid, mass transfer such as intramolecular diffusion (diffusion of the adsorbate both into the adsorbed phase and into the pore), and, finally, adsorption of the species according to physical or chemical mechanisms at the active sites of the solid [54]. Thus, in this paper, the kinetic expression covers the various stages of the adsorption process, and the slowest stage determines the observed kinetics of the entire process. Comparing the UV-VIS spectrum of silver nanoparticles (clearly spherical nanoparticles with sizes between 5 to 10 nm (Figure 1)), it can be assumed that the presence of nitrogen base influences the morphological properties of the colloidal silver. The phenomenon of the interaction of nitrogenous bases on the morphology of metallic nanoparticles was also observed previously in the literature [23]. Following the addition of the Anb to the silver nanoparticles, the surface plasmon band was shifted from 400 nm to ~500 nm, however, such a shift depends on temperature conditions. It was especially visible during the recording of UV-VIS spectra of kinetic investigation (Figure 6). This effect suggests the presence of in situ formed aggregates as a result of interaction between Anb and AgNP. The newly-generated resonance peak (above 400 nm) could be related to the dipole plasmon resonance of the aggregated surface [55], and the presence of Anb plays a role in the induction of such an effect. One explanation can be related to the presence of two exocyclic nitrogens with lone pairs of electrons participating in binding silver. To study the influences of temperature and the nanometallic phase on the adenosine adsorption rate on the activated carbon RIAA, spectrophotometric spectra were measured. Figure 6 depicts registered spectra for the adenosine adsorption process from the one- (due to the great similarity only at 10 °C) (Figure 6A) or two-component solutions at 10, 25, and 40 °C (Figure 6B–D). Based on the maximum absorption values in given moments of the experiment, the profiles of concentration (c), ~ time (t), and (c) ~ square root of time ($t^{1/2}$) were obtained. It should be explained that due to the invariability of the wavelength (259 nm) of the peak corresponding to adenosine (both in one- and two-component solutions), the kinetic curves are presented in the entire time range of the experiment. The peak corresponding to silver nanoparticles (in the two-component solution), depending on temperature, is shifted, widened, and even had its characteristics changed from mono- to bimodal, so the presented kinetic curves are truncated to approx. 1200 min. of the experiment to eliminate errors of temporary concentrations.

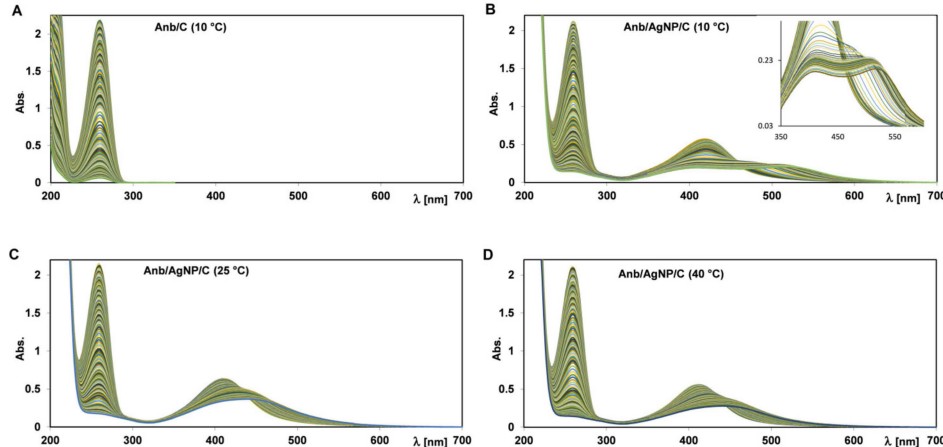

**Figure 6.** The spectrophotometric spectra collected for the adenosine adsorption process on the activated carbon RIAA from one-(**A**) or two-component solutions (**B–D**) at 10, 25, and 40 °C.

Taking into account the complexity of the experimental systems (heterogeneity of nonuniform porous carbon, macromolecular structure of adenosine, and colloidal silver phase specificity), we assumed that the kinetics of adenosine on the activated carbon RIAA from the one- and two-component solutions may be well described by the multiexponential equation. So, first, an analysis of the kinetic curves using the m-exp. equation was carried out. In Figure 7A–D the experimental data and theoretical ones for the best-fitted parame-

ters of the m-exp. equation are presented. One can see that the adenosine adsorption rate increases with the temperature (increase in kinetic energy of the adsorbate) and at higher temperatures, the equilibrium is achieved in a shorter time. This trend applies to both nitrogenous base adsorption from the one- and two-component solutions, although in the case of the latter systems, an additional significant effect of the colloidal silver phase on the adenosine adsorption rate is found. Undoubtedly, it is related to interactions between the metallic nanoparticles, adenosine, and activated carbon, visible in the absorption spectra of the metallic nanophase as a peak wavelength shift, widening, or differentiation of two separate peaks. All these changes can be explained by the decreased stability of nanoparticles, manifested in their shape and size alteration, or even the aggregation process.

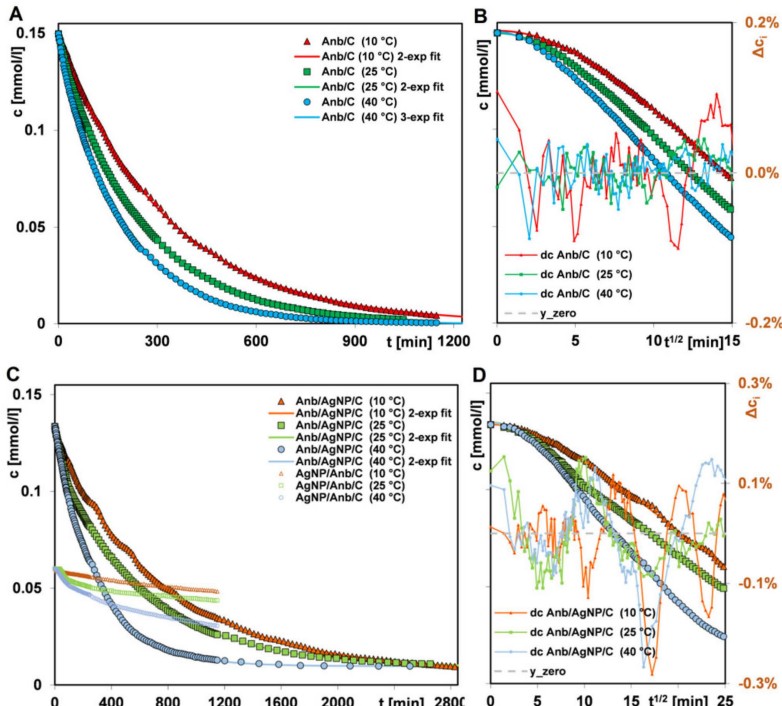

**Figure 7.** Comparison of adenosine adsorption kinetics from the one-component (**A**,**B**) and bicomponent solutions (**C**,**D**) on the activated carbon RIAA at 10, 25, and 40 °C. Comparison of silver nanoparticles adsorption kinetics from the two-component solutions on RIAA in the range time 1–1200 min at various temperatures. The lines correspond to the fitted multiexponential equation.

Based on the data in Table 3, it was found that the achievement of 50% efficiency for the adenosine adsorption from the one-component solution required times of 121, 138, and 222 min at 40, 25, and 10 °C, respectively. On average, to get the same efficiency (25, 50, or 75%) at 10 °C, as those at 40 °C, the process was prolonged by about 1.8 times (Figure 8). At 40 and 25 °C, the equilibrium was achieved after 12.5 and 16.5 h of the experiment, respectively. At 5 °C, the time of 19 h was not enough to reach an equilibrium state. A significant decrease of the adsorption rate of adenosine in the accompaniment of metallic nanoparticles was observed, though at 40 °C this effect is least meaningful (extension of time by average of 1.9 times, compared to an average of 2.4 and 2.6 at 10 and 25 °C, respectively). Taking into account that at 40 °C, the metal nanoparticles adsorption rate is greatest, one can assume a partially independent adsorption process of the metallic nanophase and adenosine. The fine nanoparticles, in comparison to larger nitrogenous base molecules, diffuse faster into the surface of activated carbon and fill the active sites, also those not available for adenosine. At 10 and 25 °C, due to the lower kinetic energy of nanoparticles, they stay in the solution longer and inhibit adenosine molecules diffusion into suitable carbon pores. Additionally, any transformation of silver nanoparticles into aggregates can lead to a blocking of access for adenosine to some carbon

pores. Graphically, the progress of adenosine adsorption from the one-component and two-component solutions at various temperatures is presented in Figure 9A–D.

**Table 3.** Comparison of times needed to achieve adsorption efficiencies at various temperatures.

| Efficiency/Temperature | Time (min) | | | | | |
|---|---|---|---|---|---|---|
| | 10 °C | | 25 °C | | 40 °C | |
| | Anb/C | Anb/AgNP/C | Anb/C | Anb/AgNP/C | Anb/C | Anb/AgNP/C |
| **25%** | 93 | 197 | 55 | 110 | 45 | 84 |
| **50%** | 222 | 555 | 138 | 387 | 121 | 228 |
| **75%** | 470 | 1200 | 296 | 920 | 258 | 478 |
| **95%** | 970 | - | 666 | - | 560 | - |

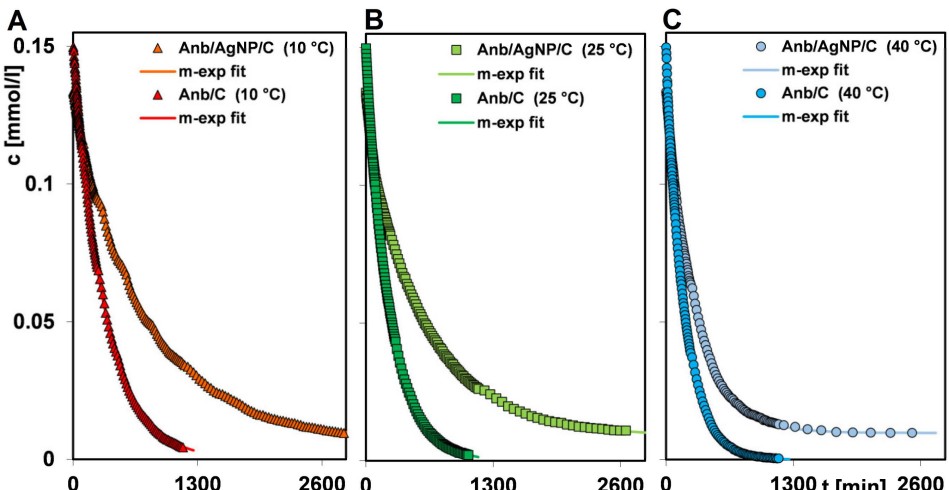

**Figure 8.** Comparison of adenosine adsorption kinetics from the one-component and bicomponent solutions on the activated carbon RIAA at 10 °C (**A**), 25 °C (**B**), and 40 °C (**C**). The lines correspond to the fitted multiexponential equation.

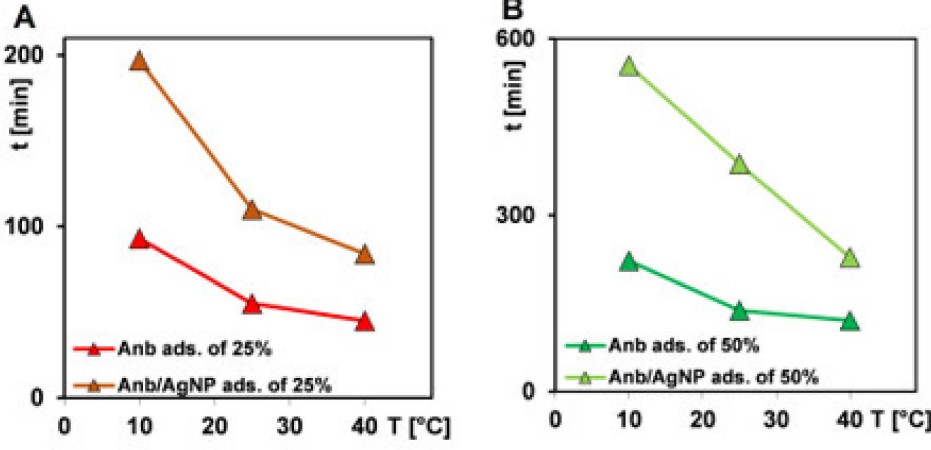

**Figure 9.** *Cont.*

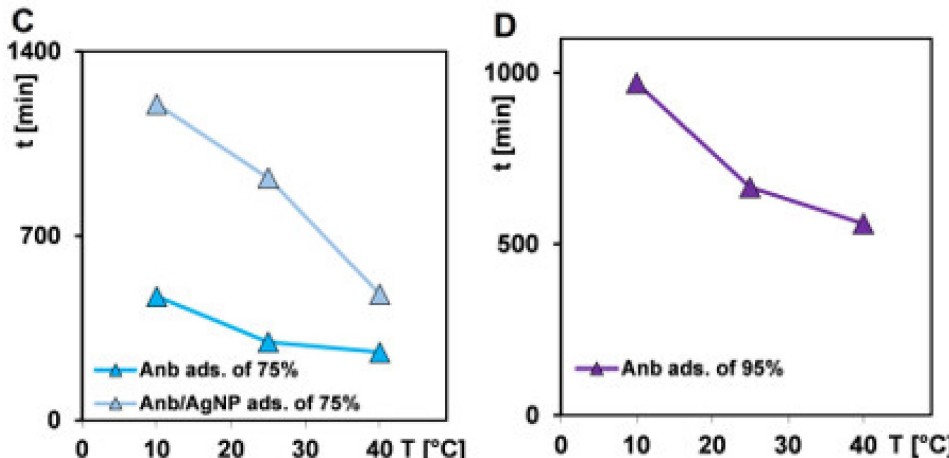

**Figure 9.** Progress of 25% (**A**), 50% (**B**), 75% (**C**), and 95% (**D**) of adenosine adsorption from the one-component and two-component solutions at various temperatures.

The mechanism of adenosine adsorption on the activated carbon RIAA was based on the π-π interactions of aromatic rings and to a lesser extent of the possible electrostatic attraction forces (between charged moieties of the solid surface and hydroxy or protonated amino groups of the adsorbate), and the hydrogen bonds. Based on the estimated values of the adsorption activation energy for the one-and two-component systems (22.56/22.7 and 14.87/13.42 kJ/mol, respectively) one can suggest that the presence of the Ag nanophase significantly weakened interactions between adenosine and activated carbon.

Based on relative standard deviations $SD(c)/c_o$, the determination of the optimal number of exponents of the multiexponential equation required to optimize kinetic data was done. For the majority of the systems, two exponents are sufficient to provide a good quality fitting, while for one system, three exponents are preferred. Graphical dependence of the fitting quality $SD(c)/c_o$ on the number of exponential terms of the m-exp. equation fitted to adenosine (single adsorbate and accompanied by silver nanoparticles) adsorption kinetics on activated carbon is drowned in Figure 10. In Table 4, parameters of considered semi-empirical equation are presented: the logarithm of rate constant $\log k_i$; the adsorption half-time $t_{0.5i}$ for the respective equation term ($t_{0.5} = (\ln 2)/k_i$); the numerically calculated the average adsorption halftime $t_{0.5}$, and the relative uptake $u_{eq}$ ($u_{eq} = 1 - \frac{c_{eq}}{c_0}$).

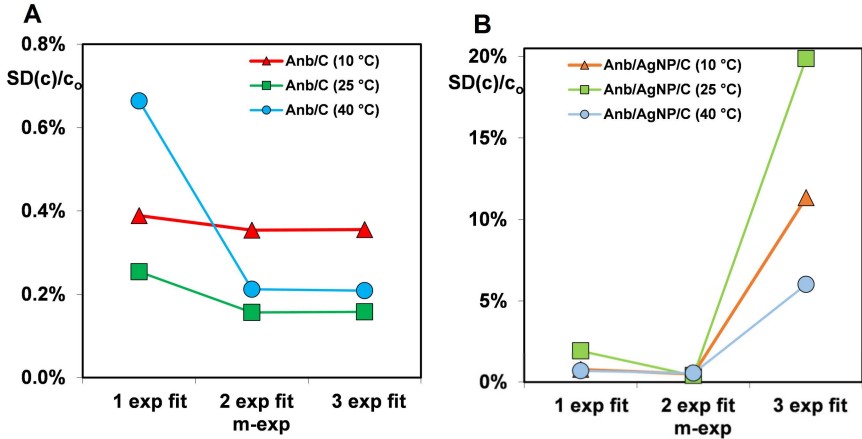

**Figure 10.** Dependence of standard deviations of relative concentration $SD(c)/c_o$ on the number of exponential terms of the multiexponential equation in fitting experimental adsorption of the adenosine from the one- (**A**) and two-component (**B**) solutions on the activated carbon RIAA.

**Table 4.** Comparison of the parameters of various kinetic models and equations.

| System | m-exp | FOE | SOE | MOE | f-FOE | PDM (a) | PDM (b) |
|---|---|---|---|---|---|---|---|
| **Anb/C (10 °C)** | 0.356% | 0.385% | 6.58% | 0.369% | 0.366% | 2.90% | 0.501% |
| **Anb/C (25 °C)** | 0.157% | 0.464% | 6.30% | 0.344% | 0.287% | 2.27% | 0.269% |
| **Anb/C (40 °C)** | 0.209% | 0.661% | 6.47% | 0.483% | 0.403% | 2.11% | 0.358% |
| **Anb/AgNP/C (10 °C)** | 0.500% | 0.778% | 4.17% | 0.559% | 0.505% | 0.680% | 0.552% |
| **Anb/AgNP/C (25 °C)** | 0.415% | 1.93% | 2.42% | 1.18% | 0.797% | 1.11% | 0.674% |
| **Anb/AgNP/C (40 °C)** | 0.556% | 0.699% | 3.94% | 0.656% | 0.623% | 1.17% | 0.754% |
| **average** | 0.366% | 0.820% | 4.98% | 0.599% | 0.497% | 1.71% | 0.518% |

Generally, the average halftimes and rate constants confirm that the process of adenosine adsorption on carbon is faster along with a temperature rise. The parameters for kinetics of adenosine adsorption from two-component solutions evidence that the process runs at a lower rate compared to the systems without the metallic phase. Analyzing relative adenosine uptakes $u_{eq}$ (1 and 0.93–0.95 for one-and two-component solutions, respectively) one can notice also the negative influence of the coadsorbate presence on adsorption in the quantitative aspect. In Figure 11, the distribution of adsorption halftimes and rate coefficients as a relative contribution of slow and fast kinetic terms in the m-exp. equation is shown. The broader distribution of the parameters means the greater differentiation of successive stages rates of the process. Faster kinetics at 40 °C corresponds to a greater contribution of shorter halftimes, $t_{05}$, and higher rate constants, $k_i$, in comparison to those at other temperatures. Similarly, differentiation in the distribution of rate parameters for the adenosine adsorption from the one- and two-component solutions at a given temperature is noticeable. Due to the multiexponential equation enabling to obtain the average adsorption half-time $t_{0.5}$ numerically, this parameter can be used to find temperature dependence of adsorption kinetics. In Figure 12, the dependences of the logarithm of average halftimes on $1/T$ for the investigated systems at various temperatures are drawn.

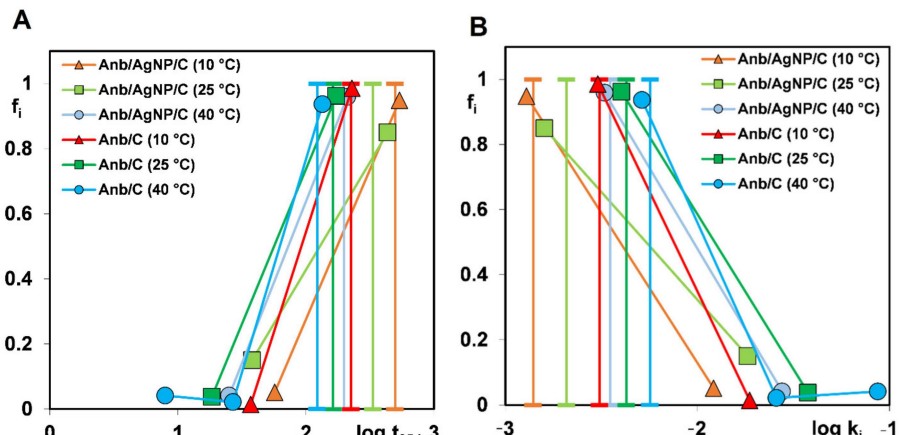

**Figure 11.** Comparison of distributions of adsorption half-times $t_{05,i}$, (**A**) and $k_i$ (**B**) for adsorption of adenosine from the one- and two-component solutions on the activated carbon RIAA. Parameters $t_{05,i}$ and $k_i$ obtained by fitting data to the m-exp equation.

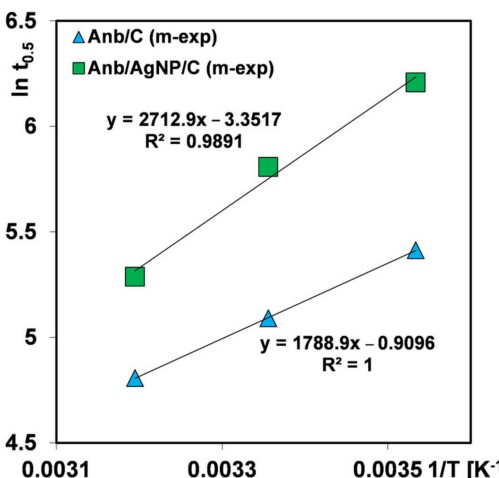

**Figure 12.** The dependences of the logarithm of average half-times on 1/T for adsorption of adenosine from the one- and two-component solutions on the activated carbon RIAA. Parameter $t_{05}$ was obtained numerically from the m-exp equation.

Based on obtained dependences and relationship (12) developed by Marczewski [56], thermodynamic constants were determined. The values of adsorption activation energy equal 14.87 and 22.56 kJ/mol for the one- and two-component systems, respectively. The significant difference between these values may be explained by interactions between the metallic nanoparticles and other system components which hinder adenosine adsorption.

$$ln\ t_{0.5} \approx lnB - ln(q_T T) + \frac{E^*}{RT} \tag{12}$$

where B is a model-dependent constant, $q_T$ is a temperature-dependent function, and E* is the activation energy. In line with our assumptions, a very good correlation between experimental and fitted by m-exp. equation data is observed. All studied systems are characterized by low values of the determination coefficient $1-R^2$ (mean $1.2 \times 10^{-5}$) and relative standard deviation $SD(c)/c_o$ (mean 0.37%), (Table S1). Moreover, the high-quality optimization can be confirmed by no systematic deviations of put-together data, (Figure 6) as curves: deviations of concertation ($\Delta c_i$)~square root of time ($t^{1/2}$), where $\Delta c_i = c_i - c_{fit}(t_i)$.

The next step of analysis of the kinetic data was an investigation of the approximation of the process using simple model-based equations as follows: first order (FOE), second order (SOE), and mixed 1.2 order (MOE) equations. None of the mentioned methods of optimization give as good results as a multiexponential equation. MOE and FOE deviations are, on average, 1.6 and 2.2 times worse than m-exp. ones (mean $SD(c)/c_o$: 0.6 and 0.82 vs. 0.37%). The fitting parameters for SOE are much worse (mean $SD(c)/c_o$: 4.5%) and are of systematic features.

Since the FOE, SOE, and MOE equations relate to Langmuir kinetics, we also tried to use a fractal-like MOE equation, derived from MOE. This equation reflects a conception of fractality, so it seems more suitable for describing the kinetics of heterogeneous systems. The f-MOE equation includes in its assumptions a certain distribution of adsorption rates like in the case of the multiexponential equation. In the process of optimization using the f-MOE equation for all the studied systems, negative values of $f_2$ were obtained. Therefore, the f-MOE was reduced to the simple form of the f-FOE equation. In Figure 13, the dependences of various rate parameters from the f-FOE equation on 1/T for the adenosine adsorption from the one- and two-component solutions on the activated carbon RIAA are presented. Logarithms of ($t_{0.5}$); ($k_1$); and ($k_1/T$) ~ (1/T) are characterized by ideal ($R^2 \approx 1$) or satisfied linearity ($R^2$=0.9346–0.9814), which allowed for the estimation of the adsorption activation energies. Obtained values from the given dependences equal: 14.53, 14.10, 11.63 kJ/mol and 22.63, 22.97, 20.50 kJ/mol for adenosine adsorption from the one- and two-component solutions, respectively. Thermodynamic constants, estimated based

on $t_{0.5}$ and $k_1$, are most similar and consistent with ones obtained from the m-exp equation parameters (14.87 and 22.56 kJ/mol).

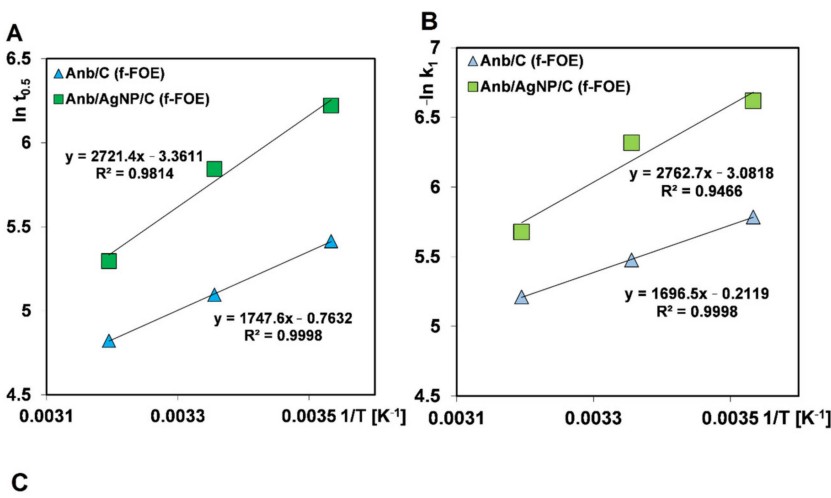

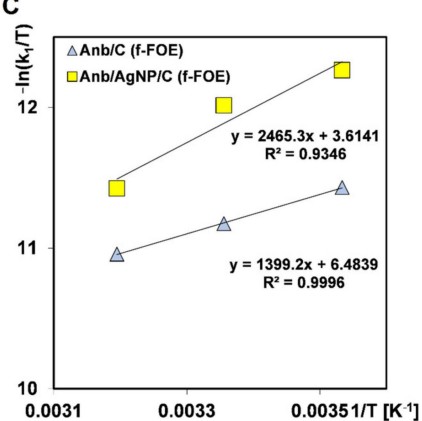

**Figure 13.** The dependences of various rate parameters: $t_{0.5}$ (**A**), $k_1$ (**B**), and $k_{1/T}$ (**C**) on $1/T$ for adsorption of adenosine from the one- and two-component solutions on the activated carbon RIAA). The rate parameters were obtained from the f-FOE equation.

The fitting quality of f-FOE is close to that of the m-exp. equation-relative deviations and are only, on average, 1.4 times (mean $SD(c)/c_o$: 0.5%) worse. For a better presentation of the fitting quality using various kinetic equations, the experimental data, fitted lines (m-exp., FOE, SOE, MOE, and f-FOE), and their deviations in Weber-Morris linear coordinates for the exemplary systems are plotted in Figure S1 (Supplementary Materials). In Table S1, the parameters of considered kinetic equations are summarized.

Finally, we analyzed whether the kinetics of adenosine adsorption on the activated carbon RIAA behaves following the intraparticle diffusion model (IDM). For this purpose, the experimental data were presented in Bangham's linear coordinates (Figure S2). However, despite obtaining high linearity of function, especially for one–component systems ($R^2$: 0.9947–0.9995) the values of slope parameters ranged from 0.7916–1.0501 instead of 0.5, which is typical for IDM. One can note that the linearity of the Bangham plot for the entire range of experimental data, mathematically is equivalent to the f-FOE kinetic equations: $\frac{c}{c_o} = 1 - \exp\left[-(kt)^s\right]$, with $s = p$ ($p$-fractality parameter) and $c_{eq}/c_o = 0$. This again confirms the high approximation of the adenosine adsorption rate to the f-FOE kinetics.

The unsuccessful results of the Bangham plot for the IDM model prompted us to skip it and use the McKay model (PDM-pore diffusion model) instead. The PDM model is a modification of IDM, giving more flexibility in the description of kinetic data by introducing additional theoretical assumptions. There are two versions of the pore diffusion model: without and with optimization of relative uptake $u_{eq}$, which are labeled as PDM (a) and PDM (b), respectively. The PDM(a) model was first used with a relatively low

quality of fitting (mean SD(c)/$c_0$: 1.71%), which shows that this model cannot describe the adenosine adsorption rate on the activated carbon RIAA. However, the treatment of $u_{eq}$ as the optimized parameter gives more successful results (mean SD(c)/$c_0$: 0.52%), though they remain worse than for the multiexponential and f-FOE equations (Table S1).

As shown in Figure S3, the difference between the fitting quality of kinetics using PDM(a) and PDM(b) is meaningful, though there are systematic deviations. The extension of the number of optimized parameters improves the fitting quality, however, one can note still some limitations of the model related to the assumption of attaining the equilibrium at a definite time. Presentation of the experimental data and fitted PDM(b) curves in logarithmic coordinates (Figure S4) let us see discrepancies between them as the line bends at the moment of reaching the equilibrium state.

## 4. Conclusions

The interaction of biomolecules such as nitrogenous base with nanosized metallic phases is a fairly effective process, which is confirmed by the morphological changes of AgNPs as a result of mutual interactions (changes in size and shapes of the nanosized silver phase). The presented research and results extend the current knowledge in the field of interactions between colloidal metallic particles and molecules of biological importance. In particular, a significant impact of these individuals was shown, which translates into morphological changes in the nanosilver phase and is visible in the values of adsorption activation energy. In addition, the temperature of the tested system was indicated as a factor differentiating the observed morphological changes. To determine the impact of these interactions, the adenosine adsorption process was studied, taking into account the kinetic mechanisms in the system with silver nanoparticles on a carbon carrier. In this paper, the kinetic expression covers the various stages of the adsorption process (mass diffusion and strict adsorption phenomena). The kinetic study of adenosine adsorption on the activated carbon RIAA in various temperature conditions revealed that at the higher temperature of the experiment, a shorter time for achievement of the given efficiency (121, 138, and 222 min at 40, 25, and 10 °C, respectively to achieve 50% efficiency) and equilibrium state (12.5, 16.5 and over 19 h at 40, 25 and 10 °C, respectively) was required. This dependence resulted mainly from an increase in the kinetic energy of the adsorbate with the raising of the temperature. In the case of two-component systems (metallic nanoparticles as an accompanying phase), the temperature effect was slightly weakened, while the time to achieve the same adenosine adsorption efficiency was significantly extended. At 40 °C the slightest competing effect of metallic nanophase on adenosine rate adsorption was observed, which can be assigned to a partially independent adsorption process of the coadsorbates (extension of time by average of 1.9 times, compared to an average 2.4 and 2.6 at 10 and 25 °C, respectively). The fine nanoparticles, in comparison to larger nitrogenous base molecules, diffuse faster into the surface activated carbon and fill the active sites, also those not available for adenosine. In lower temperatures, the transformation of silver nanoparticles into aggregate resulted in the blocking of access of adenosine to some carbon pores. Different mutual interactions between the nitrogen base and silver nanoparticles are reflected in the shifting of the surface plasmon band from 400 nm to ~500 nm depending on the temperature of the experiment. Adenosine is indicated as an induction agent of the dipole plasmon resonance of the aggregated surface due to two exocyclic nitrogens with lone pairs of electrons participating in binding silver.

The analysis of kinetic data for several equations, such as the intraparticle diffusion model, the pore diffusion model, the first-order kinetic equation, the second-order equation, the mixed 1.2-order equation, the fractal-like MOE equation, and the multiexponential equation revealed the highest quality optimization results for the multiexponential and the fractal-like MOE equations. Generally, the average halftimes and rate constants confirm that the process of adenosine adsorption on carbon is faster along with a temperature rise. Faster kinetics at 40 °C correspond to a greater contribution of shorter halftimes $t_{05,i}$ and higher rate constants $k_i$ in comparison to those at other temperatures. Based on obtained

parameters from the multiexponential, the adsorption activation energy was estimated. Much higher values for the two-component than for the one-component system (22.56 and 14.87 kJ/mol, respectively) were found. The same trend in values of the adsorption activation energy (on average 22.7 and 13.42 kJ/mol, respectively) determined from parameters of the fractal-like MOE equations was observed. The differences between values of the thermodynamic constant were a consequence of interactions between the metallic nanoparticles and other system components.

A satisfying performance of the pore diffusion model (PDM (b)) in the description of kinetic dependences proves that diffusion of the adenosine molecules in a pore structure towards the activated carbon surface is one of the main limiting factors which affects the overall process.

**Supplementary Materials:** The following supporting information can be downloaded at: https://www.mdpi.com/article/10.3390/app13063696/s1. Figure S1: Comparison of the kinetics of adenosine adsorption from the one- (A) and two-component (B) solutions at 10 °C on the activated carbon RIAA fitted to various equations; Figure S2: Comparison of Bangham plots for adenosine adsorption from the one- (A) and two-component (B) solutions on the activated carbon RIAA; Figure S3: Comparison of kinetics of adenosine adsorption from the one- (A) and two-component (B) solutions at 10 °C on the activated carbon RIAA fitted to McKay model; Figure S4: Comparison of adenosine adsorption kinetics from the one-component (A,B) and bi-component solutions (C,D) on the activated carbon RIAA at 10, 25, and 40 °C. The lines correspond to the fitted McKay model; Table S1: Comparison of fitting as relative deviation, SD(c)/c0 for kinetic models and equations.

**Author Contributions:** Conceptualization, M.Z.-S.; methodology, M.Z.-S. and M.B.; software, M.Z.-S. and M.B.; validation, M.Z.-S. and M.B.; formal analysis, M.Z.-S. and M.B.; investigation, M.Z.-S. and M.B.; writing—original draft preparation, M.Z.-S.; writing—review and editing, M.Z.-S. and M.B.; visualization, M.Z.-S.; supervision, M.Z.-S. All authors have read and agreed to the published version of the manuscript.

**Funding:** This research received no external funding.

**Institutional Review Board Statement:** Not applicable.

**Informed Consent Statement:** Not applicable.

**Data Availability Statement:** The data are available by the corresponding author.

**Conflicts of Interest:** The authors declare no conflict of interest.

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
