# Peer review of "Nitrogenous Bases in Relation to the Colloidal Silver Phase: Adsorption Kinetic, and Morphology Investigation"

_applsci, doi:10.3390/app13063696_

Round 1

Reviewer 1 Report

The article is in general well written and well organized. The English is accurate and needs only few minor corrections. The Abstract shows the main goal of the work, in the fields of great interest nowadays of inorganic (nanoparticles) and organic moieties (biological molecules) interactions.

Considering that the work described in the present manuscript was developed and performed in this area, the importance of phosphorus as chemical element and also the importance of phosphorus containing compounds (i.e. heterocycles and organic-inorganic hybrid materials) should be better emphasized and explained. For instance, in the Introduction more papers could be cited and added to the References list, from this fields of phosphorus and nitrogen containing heterocycles and obtaining mixed dual-character compounds (organic-inorganic hybrids), as for example the work of Ilia et al. (Synthesis, Properties and Structures of Phosphorus–Nitrogen Heterocycles, Heterocycles, 2011, 83 (2), 275–291. DOI: 10.3987/REV-10-685) and Gheonea et al. (Surface Modification of SnO2 with Phosphonic Acids, Journal of Chemistry, 2017,  https://doi.org/10.1155/2017/2105938) as examples of organic-inorganic hybrids synthesis.

Phosphorus element has an important role in human body, because it is involved among others, in DNA, phospholipids, cellular membrane, ATP molecule, and so on. Therefore, its great importance should be better explained and discussed by citing more papers relevant to the fields of heterocycles and organic-inorganic hybrids (especially more recent papers, from the last decade).

The adsorption of heterocycles compounds is just briefly mentioned in the Introduction. More information could be added there, possibly also on the direction of adsorption / interaction of proteins and lipids in the human body (as surfactant/polyelectrolyte assembly). This will be useful for the readers, in order to understand the kinetics and the interactions mechanisms.

The Materials and Methods section offers many details and is well written.

The TGA was performed on nitrogen or in other atmosphere (for example it is possible to conduct the measurements in synthetic air – i.e. mixtures of nitrogen and oxygen)? This should be specified and the reason for choosing one or another should be explained.

The results and the discussion are described carefully, with a lot of information, which are also support for the conclusions. At the Results one detail should be added, as follows: while in the methods section, a coupled technique TEM/EDX is mentioned, it would be interesting to include, show and discuss also the EDX results. I’ve saw the TEM images, but the EDX data and graphs are missing. This will bring even more value for the article, because SEM-EDX is more common, but TEM-EDX is rather rarely used (but when used, as here, is more interesting, due to the higher resolution of TEM).

After performing those minor revisions, the manuscript can be considered for publication.

Reviewer 2 Report

The work of M. Zienkiewicz-Strzalka and Blachnio entitled “Nitrogenous bases in the relationto colloidal silver phase. Adsorption kinetic and morphology investigation” covers the preparation and characterization of different materials based on carbon, carbon and adenine bases and carbon, adenine and silver nanoclusters. They investigated mainly the adsoption processes of these materials and applied different equations and approaches to extract data about the pore composition, kinetics of the adsorption and so on. I do have a major concerning about the work associated to the presentation itself. There are nurmerous errors in the graphics and text that shall be fixed. For instance, Figure 3B is displaced and there is no possibility to observe it, the legends of the figures are missed (Figures 4, 5...), the text is not justified after page 11...

In addition, the work used a lot of graphics to explain their findings which confuses the reader. The lest important for the discussion shall be incorporated into ESI.

Author Response

Reviewer 2:

  1. The work of M. Zienkiewicz-Strzalka and Blachnio entitled “Nitrogenous bases in the relationto colloidal silver phase. Adsorption kinetic and morphology investigation” covers the preparation and characterization of different materials based on carbon, carbon and adenine bases and carbon, adenine and silver nanoclusters. They investigated mainly the adsoption processes of these materials and applied different equations and approaches to extract data about the pore composition, kinetics of the adsorption and so on. I do have a major concerning about the work associated to the presentation itself. There are nurmerous errors in the graphics and text that shall be fixed. For instance, Figure 3B is displaced and there is no possibility to observe it, the legends of the figures are missed (Figures 4, 5...), the text is not justified after page 11...Our Answer: Thank you for your comments. Indeed, the work available in the system has been unformatted and its editing presentation is faulty. This was absolutely not the intention of the authors. We prepared our manuscript very carefully, but not in a template. The automatic transformation of the work into a template deprived it of the correct visual aspect. It is a pity that the Editor did not return the work for correction before sending it to the Reviewers. (We have a lesson for the future). All the more thank You for the effort of reviewing the work with visible editing changes. We hope that the current version will satisfy the Reviewer.

  1. In addition, the work used a lot of graphics to explain their findings which confuses the reader. The lest important for the discussion shall be incorporated into ESI.Our Answer Thank You for your findings. Figures from 14-17, and Table 4 were transferred to the Supplementary Materials file for better visibility of the manuscript.

Reviewer 3 Report

The manuscript "Nitrogenous bases in the relation to colloidal silver phase. Adsorption kinetic and morphology investigation" is an interesting work that includes a lot of work and contains many data. The work fits in your journal rigors and  I recommend its publication

1. Introduction: The research questions are clearly defined in the state-of- the art context;

2. Materials and Methods: References must be given for Horvath- Kawazoe (HK) method (line 124), non-local density functional theory (NLDFT) method (line  125), Barrett, Joyner, and Halenda (BJH) procedure(line 128), pseudo-first order equation in Table 1.

3. Results and Discussion: This  part seems  more a description of figures. The authors do not give enough insight into the discussion of the results. Therefore, the experimental data must much, much better interpreted and explained. The practical relevance of the research should be also point out.

4. Conclusions must show how that research moves scientific knowledge forward in the papers’ topic

5. Language is very, very bad and this make  difficult to understand its scientific contribution

Reviewer 4 Report

The manuscript presents a comprehensive study concerning the adsorption of adenosine on porous activated carbon in the presence of  colloidal silver solution. The paper is interesting and well wrtitten. I have the following remarks:

1) The manuscirpt should be revised in terms of a few editing issues like:

- "2.1. . Materials" (lines 83, 118, 133, etc.)

- subsection numbering in section 3

- invisible Fig.3 C

- rows in tables 3 and 4

- captions in table 4 and Fig. 17

2) Line 194-197:

- 260 cm3/g STP not m2/g

- isotherms are very steep in this initial region and it is hardly to believe that "for all cases, the quantities of adsorbed nitrogen at 0.03 p/po" are almos the same. This statement implies the next one: "This suggests that the micropores of the system remain open and do not play a major role ..." which seems to be inconsistent with follow-up observations (lines 219-223).

3) The term "adsorption kinetics" used in the manuscript is acceptable in this context, however one should keep in mind that what is really observed in experiments is a kinetics of mass transfer in the pores of adsorbent. Strictly saying "adsorption kinetics" refers to the usually instantaneous moment of interaction of molecules with a solid surface.

4) The above distinction is important because the last phrase in  Conclusions (lines: 493-496) is a rather risky statement. In my opinion diffusion of large molecules in a pore structure towards adsorbent surface should be seen as one of main limiting factors which affect the overall process of adenosine adsorption. In spite of poor preformance of diffusion models.

And off the record: Isn't the image in Fig. 5E similar to the tiger from Winnie the Pooh?

Reviewer 5 Report

I did not find any serious drawbacks in the MS. I recommend to accept it in its current form.

Author Response

  • I did not find any serious drawbacks in the MS. I recommend to accept it in its current form.

Our Answer: Thank You for Reviewer’s opinion.